# HOX epimutations driven by maternal SMCHD1/ LRIF1 haploinsufficiency trigger homeotic transformations in genetically wildtype offspring

Shifeng Xue [1,2✉], Thanh Thao Nguyen Ly[1,2], Raunak S. Vijayakar [3], Jingyi Chen[1], Joel Ng[1], Ajay S. Mathuru [2,3,4], Frederique Magdinier[5] & Bruno Reversade [2,6,7,8✉]

The body plan of animals is laid out by an evolutionary-conserved HOX code which is colinearly transcribed after zygotic genome activation (ZGA). Here we report that SMCHD1, a chromatin-modifying enzyme needed for X-inactivation in mammals, is maternally required for timely HOX expression. Using zebrafish and mouse *Smchd1* knockout animals, we demonstrate that Smchd1 haplo-insufficiency brings about precocious and ectopic HOX transcription during oogenesis and embryogenesis. Unexpectedly, wild-type offspring born to heterozygous knockout zebrafish *smchd1* mothers exhibited patent vertebrate patterning defects. The loss of maternal Smchd1 was accompanied by HOX epi-mutations driven by aberrant DNA methylation. We further show that this regulation is mediated by Lrif1, a direct interacting partner of Smchd1, whose knockout in zebrafish phenocopies that of Smchd1. Rather than being a short-lived maternal effect, HOX mis-regulation is stably inherited through cell divisions and persists in cultured fibroblasts derived from FSHD2 patients haploinsufficient for SMCHD1. We conclude that maternal SMCHD1/LRIF1 sets up an epigenetic state in the HOX loci that can only be reset in the germline. Such an unusual intergenerational inheritance, whereby a phenotype can be one generation removed from its genotype, casts a new light on how unresolved Mendelian diseases may be interpreted.

[1] Department of Biological Sciences, National University of Singapore, Singapore, Singapore. [2] Institute of Molecular and Cell Biology, A*STAR, Singapore, Singapore. [3] Yale-NUS College, Singapore, Singapore. [4] Department of Physiology, School of Medicine, National University of Singapore, Singapore, Singapore. [5] Aix-Marseille University, INSERM, Marseille Medical Genetics, Marseille, France. [6] Genome Institute of Singapore, A*STAR, Singapore, Singapore. [7] Department of Paediatrics, School of Medicine, National University of Singapore, Singapore, Singapore. [8] Department of Medical Genetics, KOÇ University, Istanbul, Turkey. ✉email: dbsxues@nus.edu.sg; bruno@reversade.com

Early embryonic development is controlled by the loading of maternal factors into the oocyte during oogenesis. These RNA and proteins, also known as maternal effect genes, support embryogenesis until the zygotic genome is activated for transcription[1]. Maternal effect genes are known for a range of functions including cell cycle regulation, genomic imprinting, pluripotency and cell fate specification[2]. However their effects are usually restricted to embryogenesis and rarely long-lasting.

Structural maintenance of chromosomes flexible hinge domain containing 1 (SMCHD1, MIM614982) is an evolutionarily conserved chromatin-binding protein involved in epigenetic silencing[3]. SMCHD1 was first discovered to be critical for X-inactivation in mammals since homozygous mutant female mice die by mid-gestation while males remain healthy and fertile[4]. Subsequently SMCHD1 was also documented to regulate the expression of mono-allelically expressed and imprinted genes[5]. Germline mutations in *SMCHD1* have been implicated in two very distinct Mendelian human diseases. Haploinsufficiency of *SMCHD1* is responsible for facioscapulohumeral dystrophy type 2 (FSHD2, MIM158901), where the monoallelic depletion of SMCHD1 causes hypomethylation of the D4Z4 repeat region and subsequent expression of *DUX4* which is toxic to muscle lineages[6]. More recently, de novo missense mutations in the GHKL ATPase domain of SMCHD1 are shown to cause a striking craniofacial disorder, Bosma Arhinia Microphthalmia Syndrome (BAMS, MIM603457) which is characterized by the congenital absence of a nose and its associated structures[7,8].

SMCHD1 expression precedes fertilization and is found in developing oocytes in mice and humans[9–12]. Because *Smchd1* knockout female mice are embryonic lethal, the role of its maternal contribution before the onset of zygotic transcription has been difficult to address. To circumvent this limitation, we turned to zebrafish where the absence of imprinting and X-inactivation permits interrogation of the role of maternal *smchd1*.

Here we show that loss of *smchd1* in zebrafish causes derepression of a large number of *hox* genes resulting in aberrant skeletal patterning. Surprisingly this is caused largely by the maternal pool of Smchd1 as even wild-type offspring born to heterozygous knockout zebrafish *smchd1* mothers exhibited vertebrate patterning defects. The loss of maternal Smchd1 was accompanied by aberrant DNA methylation in the *hox* loci. This is conserved in mammals as we further show that *HOX* genes are also misregulated in *Smchd1* haploinsufficient mouse and human fibroblasts derived from FSHD2 patients. This regulation is mediated by Lrif1, a direct interacting partner of Smchd1, whose knockout in zebrafish phenocopies that of Smchd1. We conclude that maternal Smchd1/Lrif1 set up an epigenetic state in the *hox* loci that can only be reset in the germline. Such an unusual intergenerational inheritance, whereby a phenotype can be one generation away from its genotype, casts a new light on how unresolved Mendelian diseases may be interpreted.

## Results

### *MZ smchd1* fish embryos show derepression of *hox* genes.
Using CRISPR/Cas9 technology, we generated three distinct loss-of-function (LoF) *Smchd1* alleles causing out-of-frame mutations in the GHKL ATPase domain (Fig. 1a). Endogenous *smchd1* mRNA is heavily maternally contributed in zebrafish with ubiquitous distribution until gastrulation, after which it becomes restricted to the head and lateral plate mesoderm (Fig. 1b). This large maternal contribution makes zebrafish an ideal system for interrogating the role of maternal Smchd1. By RT-QPCR and whole-mount in situ hybridization, all three lines of maternal zygotic (MZ) null embryos showed barely detectable endogenous

*smchd1* mRNA at the 2-cell stage and protein at the sphere stage (Fig. 1c) indicating that the maternal pool of *smchd1* was degraded through the nonsense-mediated decay pathway. These three independent mutant lines represent the first complete protein-null *smchd1* knockouts in zebrafish.

A previous study suggested that zebrafish *smchd1* morphants or crispants display craniofacial anomalies mimicking BAMS[8]. In light of recent debates on whether F0-depleted animals can be used interchangeably for genetic mutants, we examined whether germline *smchd1* knockout zebrafish recapitulate BAMS. No craniofacial abnormalities were seen in: F1 zygotic heterozygous fish (which best recapitulate the *SMCHD1* genotypes of BAMS patients), F2 zygotic homozygous fish or even F3 MZ homozygous *smchd1*-null fish. At 3 months of age, MZ *smchd1* knockout fish are visually indistinguishable from wildtype (*wt*) counterparts (Fig. 1d), did not display any arhinia-associated phenotypes (Supplementary Fig. 1a–c) nor did they exhibit signs of impaired olfaction to ethologically salient attractive and aversive odorants (Supplementary Fig. 1d and Movie 1).

Having genetically and functionally demonstrated that loss-of-function of Smchd1 in zebrafish does not model BAMS, we sought to understand Smchd1's exact role during embryogenesis. We first performed an unbiased transcriptomic analysis of *wt* and *MZ smchd1^{lof1/lof1}* embryos at the 4- to 8-cell stage, when all RNAs in the embryo are maternally supplied, and at sphere stage after zygotic transcription has begun. Gene Ontology analysis of differentially expressed genes at the 4- to 8-cell stage revealed skeletal system development as one of the top categories (Supplementary Fig. 2a–d). Further clustering of the differentially expressed genes at the 4- to 8-cell stage disclosed 2 clusters of genes which were upregulated in the mutant embryos (Fig. 2a). Among these, cluster 5 showed a significant enrichment of genes involved in skeletal system development (Fig. 2b). On closer examination, a conspicuous upregulation of a large number of *hox* genes was recorded at both stages (Fig. 2c, d, Supplementary Fig. 2e, f, 3a, Data 2). Hox genes serve as master regulators of anterior posterior (AP) patterning in the embryo and their expression boundaries are tightly controlled to specify the identity of body parts[13]. At 11-somites when most *hox* genes are highly expressed, qRT-PCR analysis found that a series of *hox* transcripts were persistently upregulated (Fig. 2e). For instance the expression of *hoxb2a* is normally restricted to rhombomere 3 and 4 at 10-somites, but appeared to be extended into rhombomere 5 in MZ *smchd1^{lof1}* embryos (Fig. 2f, Supplementary Fig. 3b, c). There was also an anterior expansion of the *hoxc10a* expression boundary (Fig. 2f). These data indicate that loss of SMCHD1 is sufficient to de-repress a vast number of *hox* genes in early development.

### *MZ smchd1* fish display vertebral patterning defects.
Zebrafish has undergone a partial genome duplication and carries seven *hox* gene clusters, instead of four in mammals and one in *Drosophila*[14]. Surprisingly adult MZ *smchd1^{lof1/lof1}* fish showed patent vertebral patterning defects consistent with homeotic transformations. The observed phenotype (Fig. 2g, h) consisted of a reduction in the number of ribs, caudal vertebrae and occasionally of supraneural vertebrae (Supp. Fig. 4a–d). The future reduction of vertebrae could already be documented in 48 hpf embryos in reduced somite numbers as marked by *xirp2a* (Supplementary Fig. 4e, f). Our results show that the de-repression of multiple *hox* genes caused by the absence of *smchd1* suffices to trigger vertebral patterning defects in zebrafish.

Changes in vertebral numbers have been previously documented with manipulations of the *mir196* family of microRNAs in mouse and zebrafish[15,16]. *mir196* genes are embedded within the

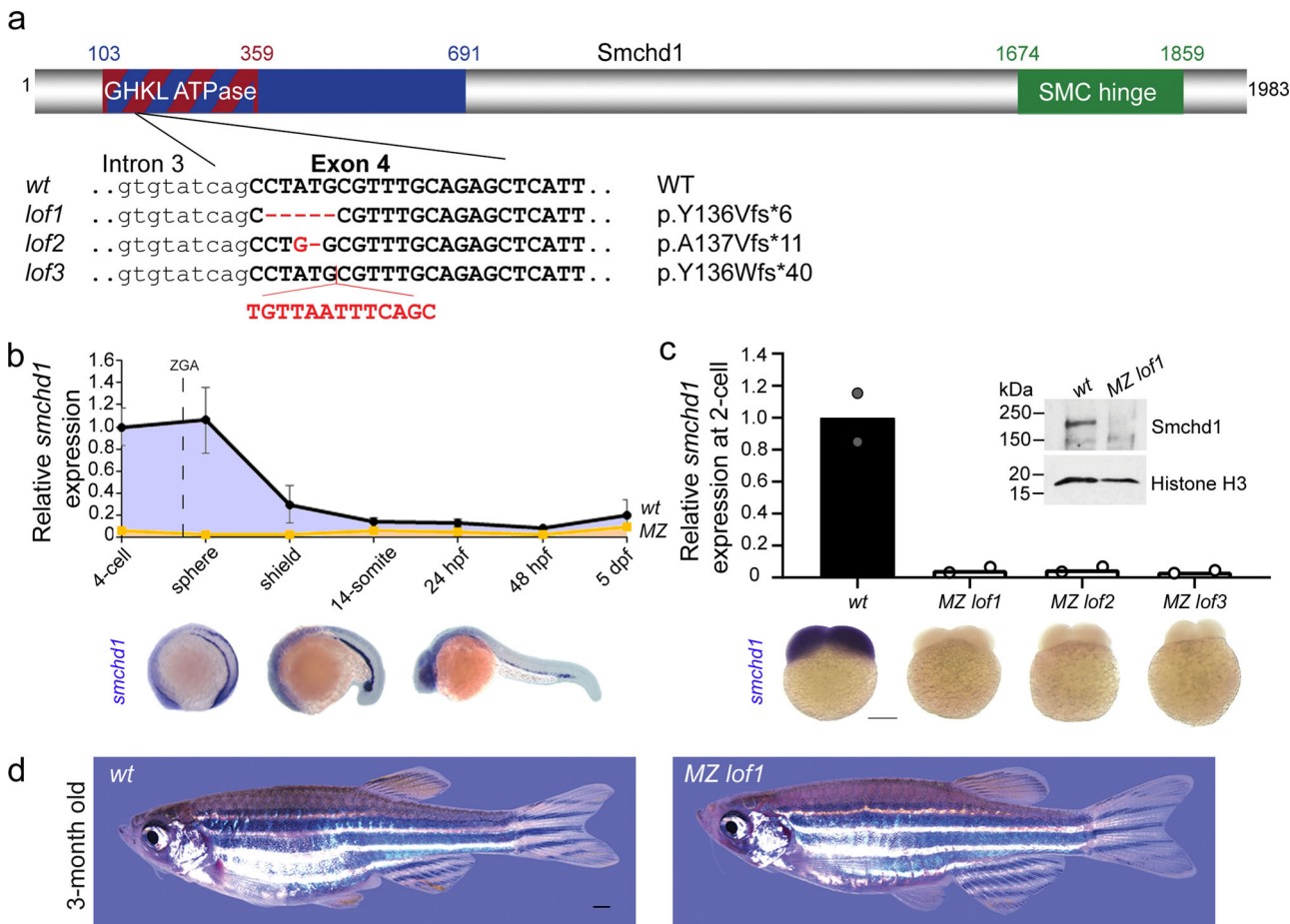

**Fig. 1 Generation of *MZ smchd1* null zebrafish. a** Three loss-of-function (LoF) alleles in zebrafish were generated by CRISPR/Cas9 injections targeting *smchd1*. **b** Endogenous spatial-temporal expression pattern of *smchd1* in zebrafish embryos as determined by qPCR and WISH. *n* = 3 biological samples (qPCR), 3 independent experiments with at least 20 embryos each (WISH). Data are presented as mean values ± SD. **c** Maternal zygotic (MZ) *smchd1^lof^* embryos lack *smchd1* mRNA (qPCR and WISH) and protein (Western blot), suggestive of total nonsense-mediated decay. Scale bar = 200 μm. *n* = 2 biological samples (qPCR, Western blot), 3 independent experiments with at least 20 embryos each (WISH). **d** *MZ smchd1^lof^* zebrafish are viable and fertile, and do not model arhinia at 3 months. Scale bar = 1 mm.

*Hox* loci and act to post-transcriptionally repress the expression of neighbouring *Hox* genes. Expression of mature *mir196* paralogs was not significantly changed in *smchd1^lof1^* oocytes but was increased approximately 1.5 fold at 11-somites, suggesting that it may also participate in the general dysregulation of the *hox* transcripts (Supplementary Fig. 3d, e). As *mir196*-induced vertebral reductions were previously achieved with an estimated 10–20 fold *mir196* over-expression, we surmise that the modest *mir196* upregulation may not be a main driver of the phenotype in our *smchd1* knockout fish.

**Skeletal phenotype is caused by maternal Smchd1 haploinsufficiency.** To assess which of the maternal or zygotic pool of *Smchd1* is responsible for proper AP axial patterning, the skeletons of maternal null *smchd1^lof1^* and zygotic null *smchd1^lof1^* were compared. Both displayed equivalent reductions in the number of vertebrae as *MZ smchd1^lof1/lof1^* (Fig. 3a, b, Supp. Fig. 4a–d) suggesting that both sources may be required. By analysing the levels of *smchd1* in paternally- or maternally-deleted embryos, we found that most, if not all, endogenous *smchd1* up to gastrulation is deposited by the mother (Fig. 3c). This led us to hypothesize that the observed vertebral defects in zygotic null *smchd1^lof1/lof1^* may be the result of insufficient maternal *smchd1* deposition in the egg rather than from a

subsequent loss of zygotic *smchd1*. To answer this, we turned our attention to genetically wildtype *smchd1^+/+^* siblings from the same brood as zygotic null *smchd1^lof1/lof1^* fish, born from the incross of heterozygous *smchd1^lof1/+^* parents. To our surprise, these fish displayed identical vertebral defects to that of zygotic null fish despite being genetically wildtype. We further generated a special clutch of *smchd1^+/+^* offspring resulting from the mating of heterozygous *smchd1^lof1/+^* mutant mothers to outbred wildtype *smchd1^+/+^* males. These fish also displayed reduced vertebral numbers (Fig. 3a, b, Supplementary Fig. 4a–d). These results demonstrate that *Smchd1* is a gene whose fractional reduction during oogenesis can alter the body plan of the future embryo irrespective of its zygotic genotype. This was surprising as the Hox code was not known to be regulated by the maternal genome. Notably, an in-cross of these phenotypically abnormal *smchd1^+/+^* fish produced offspring that were completely normal with a canonical number of vertebrae (Fig. 3a, b, Supplementary Fig. 4a–d). This rescue indicates that no trans-generational inheritance takes place and that once *smchd1* is reset to wildtype levels in the maternal germline, a proper AP patterning ensues in the following generation.

Zebrafish embryos are transcriptionally silent until the 1000-cell stage[1]. To determine whether the observed upregulation of *hox* expression in *smchd1* mutant embryos precedes fertilization or is the result of early zygotic genome activation (ZGA),

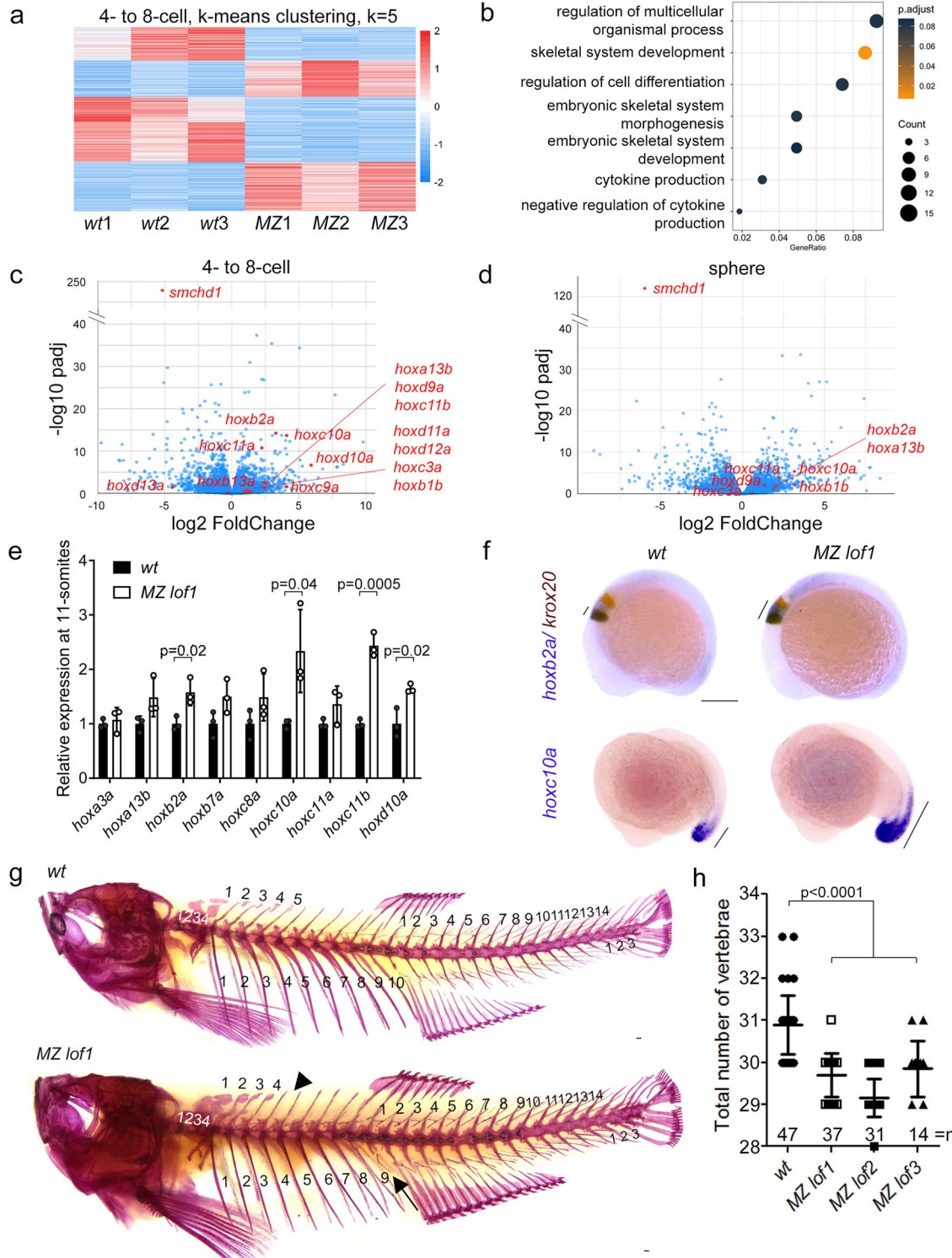

**Fig. 2 *MZ smchd1* fish embryos show derepression of *hox* genes and vertebral patterning defects. a** K-means clustering of differentially expressed genes at 4-8-cell stage. **b** Gene ontology: biological process (GO:BP) enrichment of cluster 5 from K-means clustering at 4-8-cell stage. RNA-seq of *wt* and *MZ lof1* embryos show changes in *hox* gene expression at both 4-8-cell stage (**c**) and sphere stage (**d**). **e** *hox* genes show persistent upregulation at 11-somites. $n = 3$ biological samples, *p* values were calculated by 2-tailed unpaired Student's *t* test. **f** *hoxb2a* shows expanded expression boundary at 10-somites (top) and *hoxc10a* shows anteriorly expanded expression at 17-somites (bottom). *krox20* marks rhombomeres 3 and 5. **g–h** Adult fish show defects in axial skeletal patterning, including loss of a rib (arrow) and loss of a supraneural vertebra (arrowhead). Scale bar = 200 μm. *P* values were calculated by the Kruskal-Wallis test followed by Dunn's Multiple Comparison Test. Bars without *p* values are not significant. All data are presented as mean values ± SD.

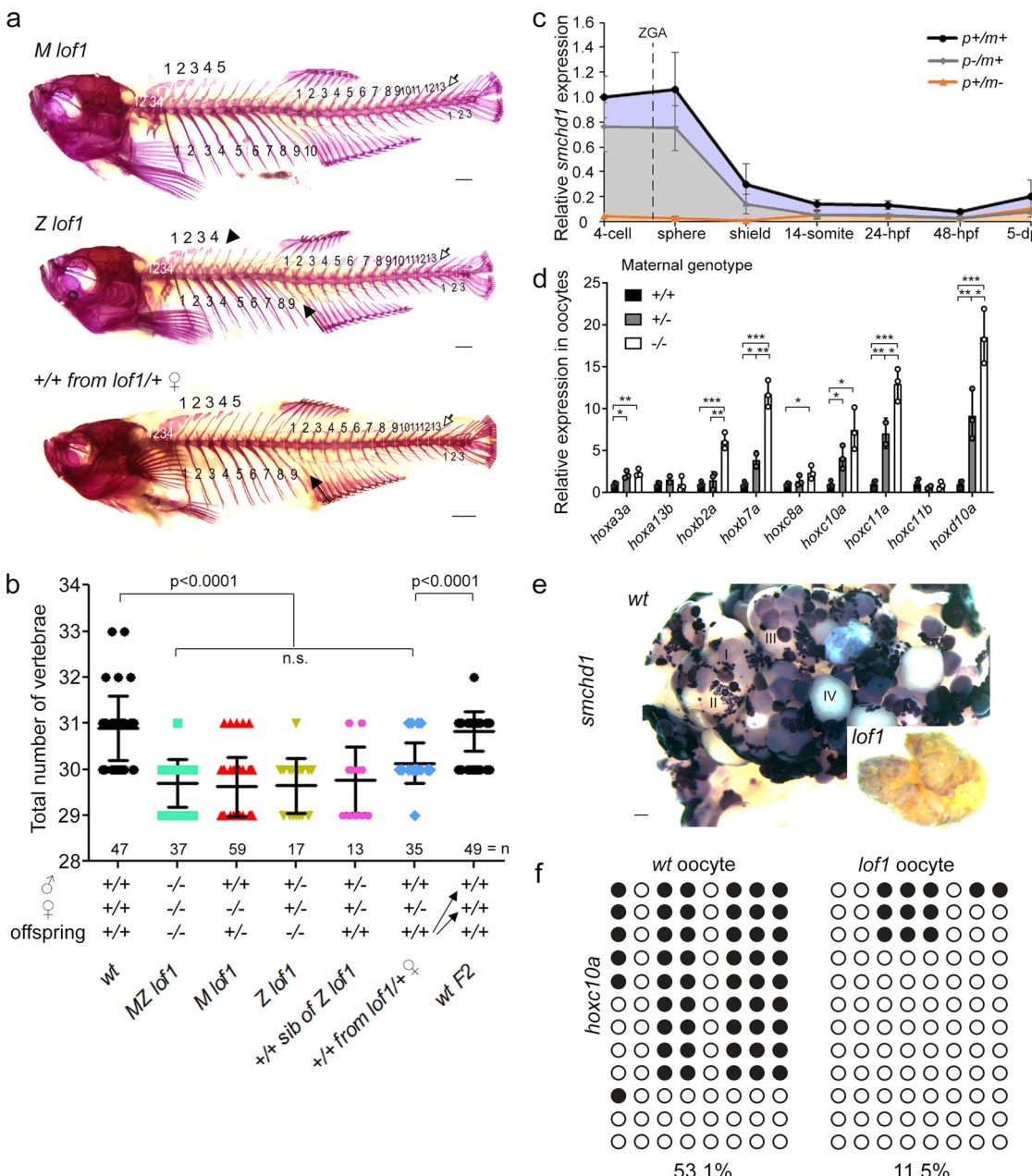

**Fig. 3 Skeletal phenotype is caused by maternal Smchd1 haploinsufficiency. a, b** Skeletal preparations of fish of different genotypes. Phenotypes include loss of a rib (filled arrow), loss of a supraneural vertebra (arrowhead) and loss of a caudal vertebra (open arrow) Scale bar = 1 mm. *P* values were calculated by the Kruskal-Wallis test followed by Dunn's Multiple Comparison Test. **c** qPCR shows that there is little to no zygotic *smchd1* expression. p: paternal allele; m: maternal allele. *n* = 3 biological samples. **d** qPCR shows that *hox* derepression is already observed in unfertilized oocytes and that oocytes from ± females show intermediate level of *hox* mis-expression. *n* = 3 biological samples, *p* values were calculated by 2-tailed unpaired Student's *t* test. **p* < 0.05, ***p* < 0.01, ****p* < 0.001. All data are presented as mean values ± SD. **e** In situ hybridization of *smchd1* shows that *smchd1* is expressed in developing oocytes. Roman numerals denote stages of oocyte development. Scale bar = 200 μm. **f** Bisulfite sequencing in oocytes shows hypomethylation of *hoxc10a* locus when *smchd1* is knocked out.

unfertilized oocytes of different *smchd1* genotypes were isolated. The levels of various *hox* genes were inversely correlated to the number of wildtype *smchd1* alleles available during oogenesis: the highest precocious *hox* expression was seen in *smchd1*^lof1 oocytes produced by homozygous *smchd1*^lof1/lof1 mothers while oocytes produced by heterozygous *smchd1*^+/lof1 mothers showed intermediate levels relative to wildtype oocytes (Fig. 3d). These results lend support to the observed maternal *smchd1* haploinsufficiency and correlate with the abundant *smchd1* expression in developing oocytes from the earliest stages of oogenesis (Fig. 3e). Our results point to a very early defect during gametogenesis in the maternal germline, where *smchd1* depletion in transcriptionally-active immature oocytes is sufficient to cause *hox* de-repression. In mammals, Smchd1 plays a role in maintaining DNA methylation on the inactive X chromosome and other imprinted genes[4,5,17]. Consistent with being hypomethylated, *smchd1*^lof1/lof1 zebrafish were less susceptible to the effects of the methylation inhibitor 5-Aza-2′-deoxycytidine (5-Aza-dC) (Supplementary Fig. 5a, b). Hox genes undergo dynamic DNA methylation changes during early embryogenesis where their methylation levels decrease dramatically from oocyte to ZGA[18,19] (Supp. Fig. 5c). In line with this, we could document the *hoxc10a* locus was markedly de-

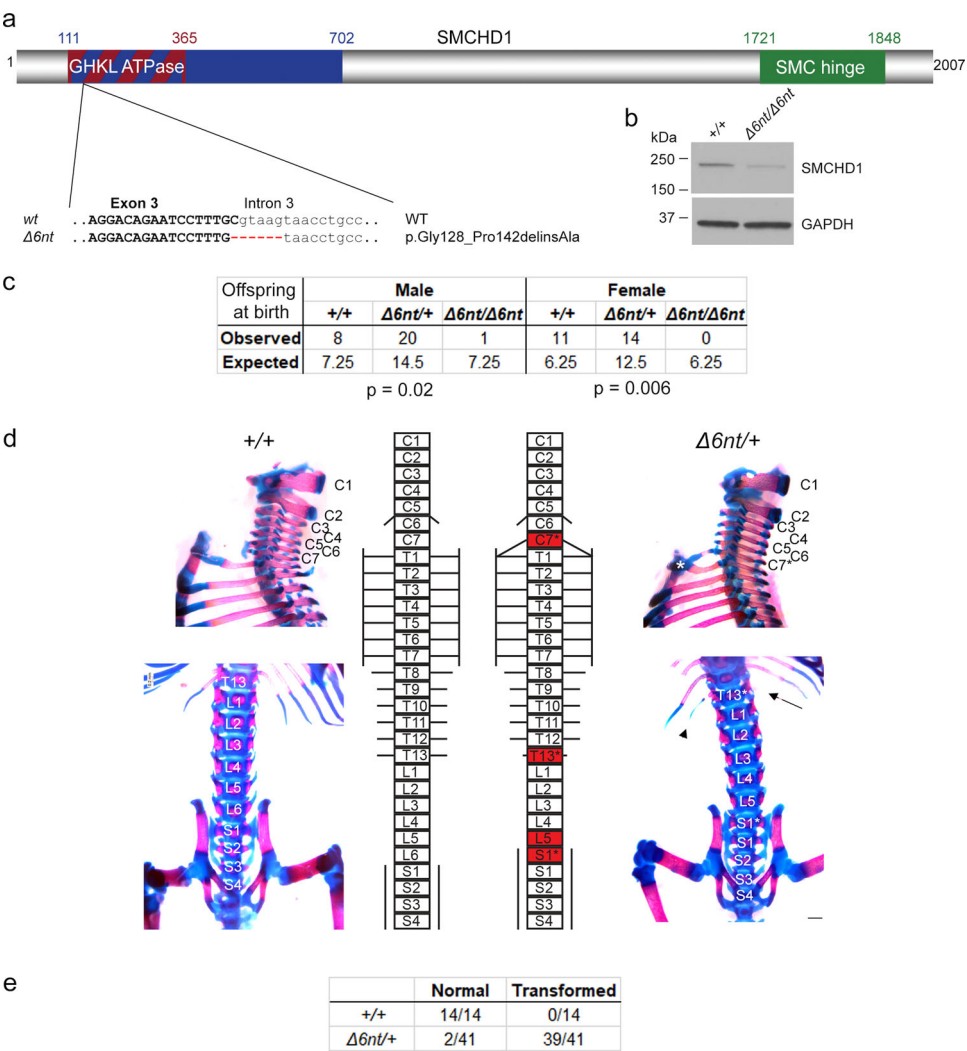

**Fig. 4 Smchd1 haploinsufficiency in mouse also causes homeotic transformations. a** A new *Smchd1* loss-of-function allele was generated using CRISPR/Cas9. This produced a 6 nt deletion including the splice donor site of intron 3. This resulted in a number of aberrantly spliced transcripts. The protein variation listed is the smallest in-frame mutation detected, a 14 amino-acid deletion including the catalytic residue of the GHKL ATPase domain p.E147. **b** Western blot from heads of E10.5 embryos show a markedly reduced level of SMCHD1 protein in the *Δ6nt/Δ6nt*, which runs slightly faster than wildtype SMCHD1. *n* = at least 5 embryos per genotype. **c** Observed and expected numbers of pups born of each genotype. Axial skeletal patterning analysis of +/+ and *Δ6nt/+* animals. Homeotic transformations observed in the animals are marked in red in the schematic (**d**). Transformations include C7 → T1, T13 → L1 and L6 → S1. C7 rib is labelled with a *, the missing T13 rib is marked with an arrow and the hypomorphic T13 rib is labelled with an arrowhead. Number of animals analysed with each phenotype are in (**e**). Animals showing any of the homeotic transformations are considered as transformed. Most *Δ6nt/+* animals show all of the homeotic transformations marked in (**d**). Scale bar = 500 μm.

methylated in *smchd1*[lof1] oocytes as compared to *wt* oocytes (Fig. 3f, Supplementary Fig. 5d, e), suggesting that Smchd1 helps maintain DNA methylation during oogenesis.

**Smchd1 haploinsufficiency in mouse causes homeotic transformations**. To ensure that this inter-generational inheritance also holds true in mammals, we sought to reexamine the effects of the loss of SMCHD1 in mice. We generated a new *Smchd1* knockout allele in the C57BL/6 background using CRISPR/Cas9 (*Smchd1*[Δ6nt]). The selected germline c.603_603 + 5del mutation deletes 6 nucleotides including the splice donor of intron 3. This deleterious allele results in a number of aberrantly spliced isoforms of which the smallest in-frame deletion removes 14 conserved amino acids of the GHKL ATPase domain p.Gly128_Pro142delinsAla (Fig. 4a, b) which would generate a catalytically-dead enzyme[20]. Similar to the previously generated *Smchd1* mutant mice (Smchd1[MommeD1/MommeD1])[4], homozygous

females were embryonic lethal while a sublethal phenotype was also observed in mutant males of this new knockout line (Fig. 4c). Previous studies have shown that SMCHD1 regulates long range chromatin interactions at *Hox* clusters[21]. Axial skeletal anomalies were seen in our newly created heterozygous *Smchd1*[Δ6nt] mice whereas they have only been reported in homozygous knockout males previously[21]. The observed homeotic transformations were consistent with those previously described including an ectopic rib at C7 and absence of ribs at T13. Additionally, consistent transformation of L6 to S1 was observed (Fig. 4d, e). We were unable to ascertain whether these observed axial transformations were due to a maternal or zygotic loss of gene activity as all *Smchd1* knockout females were embryonic lethal. However, this conundrum has been recently addressed by Benetti and colleagues[22] who have shown, using an oocyte-specific Cre-mediated deletion of maternal of *Smchd1*, that the maternal pool of *Smchd1* has an important role in repressing *Hox* expression and that also leads to vertebral patterning defects in pups born to

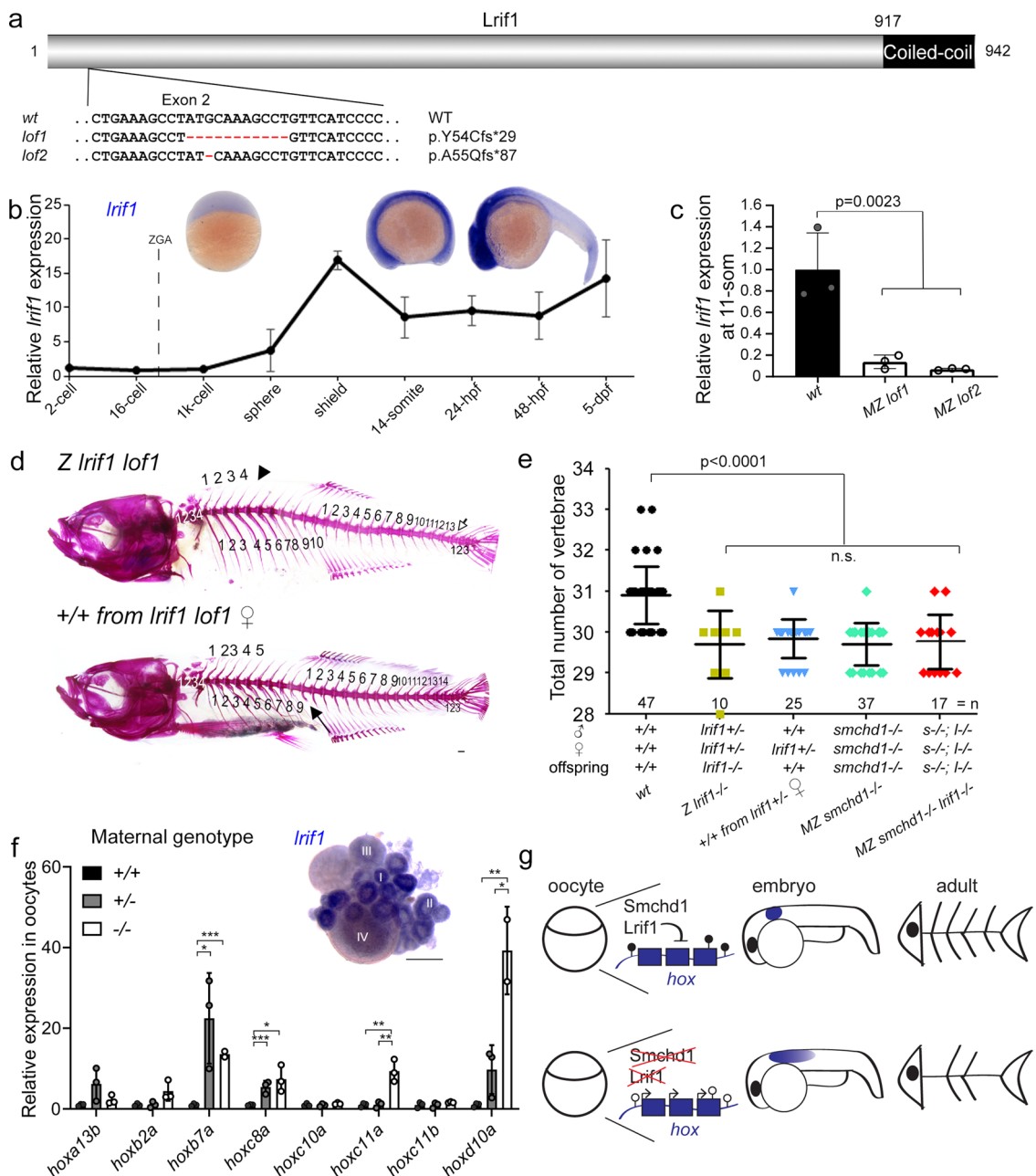

**Fig. 5 Knockout of *lrif1* phenocopies, and does not aggravate, that of *smchd1*. a** Two loss of function alleles of lrif1 were generated in zebrafish by CRISPR/Cas9 injections. **b** Endogenous spatial-temporal expression pattern of *lrif1* in zebrafish embryos as determined by qPCR and WISH. n = 3 biological samples (qPCR), 3 independent experiments with at least 20 embryos each (WISH). **c** Maternal zygotic (MZ) *lrif1lof* embryos lack *lrif1* mRNA suggestive of total nonsense-mediated decay. n = 3 biological samples. *P* values were calculated by 2-tailed unpaired Student's *t* test. **d**, **e** Axial skeletal patterning defects in adult fish. Phenotypes include loss of a rib (filled arrow), loss of a supraneural vertebra (arrowhead) and loss of a caudal vertebra (open arrow). *P* values were calculated by the Kruskal-Wallis test followed by Dunn's Multiple Comparison Test. **f** qPCR shows that *hox* derepression is detected in unfertilized oocytes from both +/− and −/− females. *P* values were calculated by 2-tailed unpaired Student's *t* test. Inset shows expression of *lrif1* in Stage I and II oocytes. Scale bars = 200 μm. n = 3 biological samples (qPCR), 5 ovaries of adult fish (WISH). **g** Model of Smchd1/Lrif1-driven transgenerational inheritance for skeletal patterning. Smchd1/ Lrif1 sets up an epigenetic state during oogenesis to ensure correct expression of *hox* genes during embryogenesis which in turn patterns the vertebrae. *p < 0.05, **p < 0.01, ***p < 0.001. All data are presented as mean values ± SD.

mothers lacking *Smchd1* only during oogenesis. These results are consistent with those made in zebrafish and confirm that Smchd1 is essential in the maternal germline, before fertilization, to control the expression of *Hox* genes in future embryos. This intergenerational inheritance documented in zebrafish which we propose is driven by *Hox* epi-mutations, may also be at play in higher vertebrate species including mammals.

**Knockout of *lrif1* phenocopies, and does not aggravate, that of *smchd1*.** Ligand-dependent nuclear receptor-interacting factor 1 (Lrif1) is the only validated protein partner of Smchd1 on chromatin[20,23]. The two nuclear proteins directly bind through SMCHD1's hinge domain and the coiled-coil domain in LRIF1 (Fig. 5a). In mammals, LRIF1 is thought to be partially responsible for recruiting SMCHD1 to the inactive X chromosome[20,23].

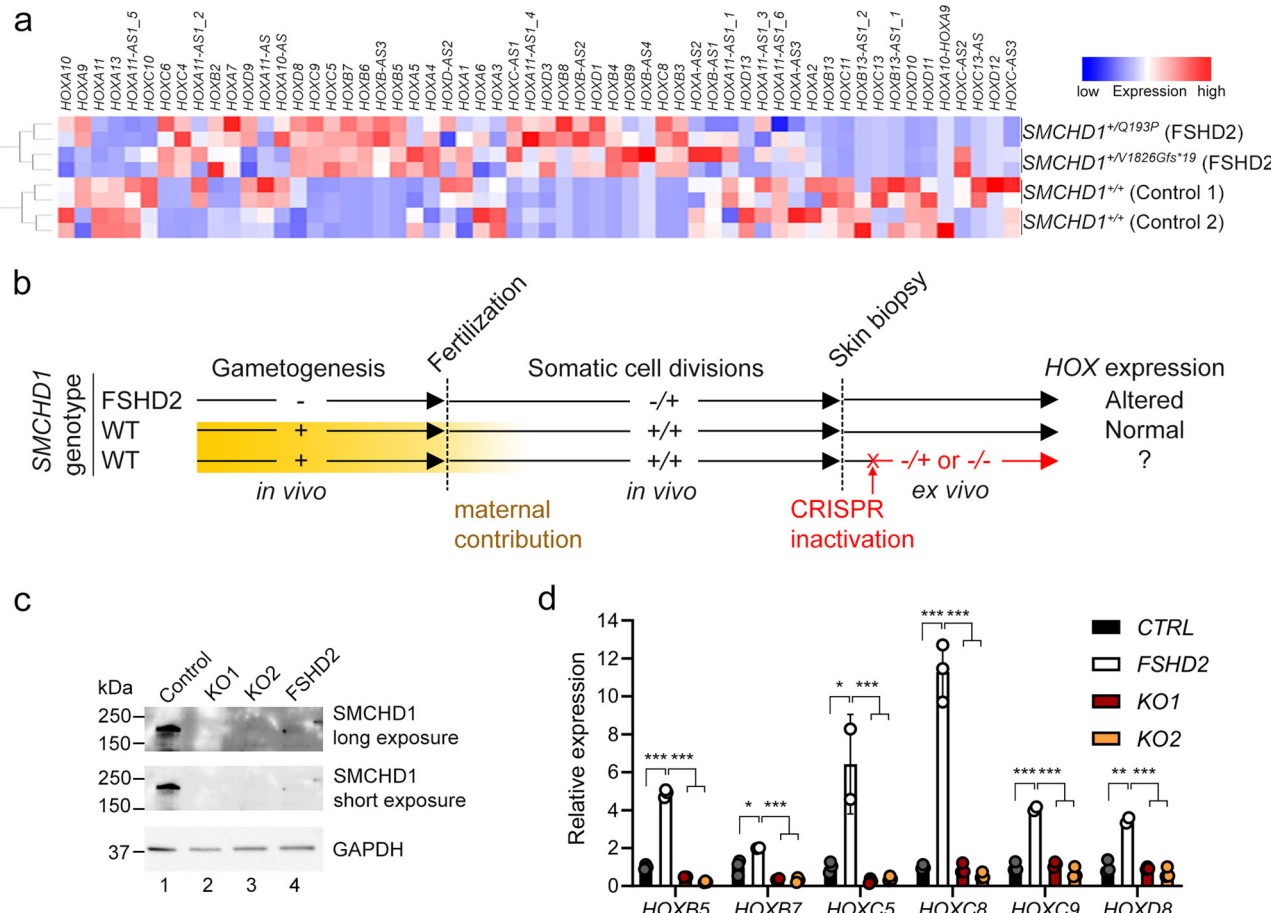

**Fig. 6 Germline, but not adult, SMCHD1 deficiency leads to *HOX* dysregulation in human cutaneous fibroblasts. a** RNA-seq performed on controls and FSHD2 patient fibroblasts reveal dysregulations of HOX expression when *SMCHD1* was mutated in the germline. **b** Schematic of experimental design to determine whether *HOX* dysregulation is the result of germline or somatic SMCHD1 haploinsufficiency. *SMCHD1* was CRISPR/ Cas9-inactivated ex vivo in control fibroblasts using two distinct guide RNAs (KO1 and KO2). Yellow box represents maternal contribution of SMCHD1 which diminishes rapidly post-fertilization. **c** Western blot in fibroblasts cell extract show successful knockout of SMCHD1, resulting in protein-null alleles. FSHD2 cells used here are +/ V1826Gfs*19. **d** qPCR shows that, unlike in FSHD2 cells, somatic knockout of *SMCHD1* in adult cells does not lead to the upregulation of *HOX*. P values were calculated by 2-tailed unpaired Student's *t* test. *$p < 0.05$, **$p < 0.01$, ***$p < 0.001$. Data are presented as mean values ± SD.

To determine if Smchd1 requires Lrif1 to control *hox* expression, we generated *lrif1* knockout zebrafish (Fig. 5a, c). Unlike *smchd1*, *lrif1* was almost exclusively zygotically produced with little to no maternal mRNA deposition (Fig. 5b). When bred to homozygosity, zygotic null *lrif1⁻/⁻* zebrafish exhibited skeletal abnormalities which were reminiscent of AP patterning defects seen in *smchd1⁻/⁻* fish. These mutant *lrif1⁻/⁻* fish had fewer precaudal and caudal vertebrae (Fig. 5d, e). Despite the absence of measurable maternal *lrif1* mRNA, we could also document that wildtype *lrif1⁺/⁺* offspring, born from *lrif1⁺/⁻* mothers crossed to outbred *wt* males, also displayed homeotic transformations (Fig. 5d, e). Likewise, a transcriptional upregulation of *hox* genes was observed in unfertilised oocytes from *lrif1⁺/⁻* and *lrif1⁻/⁻* females (Fig. 5f), suggestive of maternal haploinsufficiency. Whole-mount in situ hybridisation on ovaries showed that *lrif1* expression was detected only during the earliest Stages I and II of oogenesis (Fig. 5f). We thus speculate that the Lrif1 protein synthesized during early gametogenesis is loaded onto chromatin of maturing oocytes to actively silence *hox* genes before the ZGA. We also generated double maternal zygotic *smchd1⁻/⁻; lrif1⁻/⁻* fish, which like single knockouts, presented with comparable skeletal anomalies (Fig. 5e). This combined Smchd1/Lrif1 deficiency did not further exacerbate vertebral defects, supporting the notion that Smchd1 and Lrif1 act in the same pathway for gene

silencing. This is consistent with the two nuclear proteins working together on chromatin as a single protein complex.

**Germline, but not adult, *SMCHD1* deficiency leads to *HOX* dysregulation in human fibroblasts.** Finally, we set out to examine whether *HOX* misexpression was an ephemeral consequence of maternal *SMCHD1* depletion or rather an epigenetic anomaly that could be inherited through somatic cell divisions. To test this, we turned to SMCHD1-haploinsufficient human dermal fibroblasts taken from patients with FSHD2. Transcriptome profiling revealed a patent dysregulation of numerous *HOX* genes in two distinct FSHD2 fibroblasts as compared to those from healthy controls (Fig. 6a). To determine whether this *HOX* dysregulation is the result of haploinsufficiency of SMCHD1 during human gametogenesis and early embryogenesis, we set out to knock out *SMCHD1* in a control human fibroblast line using CRISPR/Cas9 (Fig. 6b). The somatic inactivation of *SMCHD1* resulted in protein-null cell populations (Fig. 6c) but did not lead to *HOX* dysregulation as was observed in FSHD2 fibroblasts (Fig. 6d). This is reminiscent of previous work showing that somatic SMCHD1 inactivation does not trigger hypomethylation of D4Z4[24]. Our results indicate that *HOX* misexpression caused by SMCHD1 haploinsufficiency is set up in the germline, can be stably inherited

through somatic cell divisions and cannot be recapitulated by deleting SMCHD1 in adulthood.

## Discussion

Using zebrafish, we uncovered a maternal role of *Smchd1* that was masked by the mammalian-specific processes of X-inactivation and imprinting[12]. We propose that the ancestral role of Smchd1 is to regulate developmental genes before being co-opted in mammals for imprinting. Hox genes are zygotic genes that are under multiple layers of transcriptional and post-transcriptional control[25–28]. Here we provide genetic, phenotypic and functional evidence that maternally supplied Smchd1 and Lrif1 are key in preventing premature activation of *hox* genes before fertilization by laying down an epigenetic state in the female germline (Fig. 5g). This parallels mammalian functions of maternal Trim28, which governs the epigenetic state in oocytes to ensure proper development[29]. In zebrafish, *hox* loci are distinctively occupied before ZGA by placeholder nucleosome H2AFV before resolving into bivalent chromatin marked by H3K4me3 and H3K27me3[30,31]. Future work should examine whether Lrif1/Smchd1 can recognize specific histone variants or modifications.

SMCHD1 belongs to the top 3% of genes most intolerant to mutations in humans[32,33]. Accordingly, removal of one copy of *SMCHD1* is sufficient to cause FSHD2 in humans. Recessive loss-of-function mutations in LRIF1 can also cause FSHD2[34]. Our findings document misregulation of numerous *HOX* genes in primary fibroblasts taken from FSHD2 patients. This cannot be recapitulated by the somatic inactivation of *SMCHD1*, suggesting FSHD2's aetiology, even if seen as late-onset muscular dystrophy, might have a developmental origin. This also invites the question of whether pervasive *HOX* epi-mutations, driven by insufficient SMCHD1/LRIF1 activity during gametogenesis, may contribute to the pathogenesis of FSHD2 which exhibits anterior-biased muscle degeneration. Notably, hypoplastic or absent ribs have been reported in some BAMS patients carrying heterozygous missense mutations in SMCHD1[35], drawing a possible link between SMCHD1 and anterior-posterior patterning in humans.

*smchd1* or Lrif1 haploinsufficent mothers can produce genetically-wildtype offspring that exhibit homeotic transformations. This mode of inheritance where the offspring's phenotype is one generation removed from its causal parental genotype has been referred to as inter-generational inheritance[36]. Our study may impact the interpretations of unsolved Mendelian diseases, in which the proband's phenotype may be attributed to the parent's genotype through epigenetic inheritance.

## Methods

**CRISPR-mediated zebrafish knockout**. Zebrafish were maintained and used according to the Institutional Animal Care and Use Committee of Biological Resource Centre, A*STAR, Singapore (IACUC #161172) and the National University of Singapore (IACUC #BR19-1184). Guide RNA against exon 3 of smchd1 was used with the following targeting sequence, 5′-TACGAGTATTACGCCACGCGA-3′. Guide RNA against exon 1 of lrif1 was used with the following sequence, 5′-GGGATG AACAGGCTTTGCAT-3′. gRNA was synthesized using a MEGA shortscript Kit (Thermo Fisher) according to the manufacturer's protocol and purified with an RNeasy Mini Kit (QIAGEN). Cas9 mRNA was synthesized with an mMESSAGE mMACHINE Kit (SP6) (Thermo Fisher) and a NotI-linearized zebrafish codon-optimized Cas9 construct in pCS2+. The gRNA and Cas9 mRNA were mixed together to a concentration of 375 ng/μL each, and 1 nL of RNA was injected into the yolk of 1-cell AB zebrafish embryos. Zebrafish are genotyped by PCR followed by Sanger sequencing. Primers used for genotyping are in Supplementary Data 1.

**CRISPR-mediated mouse knockout**. All mice procedures were approved by the Institutional Animal Care and Use Committee of Biological Resource Centre, A*STAR, Singapore (IACUC #201555). All mice used were on the C57BL/6 background. The Smchd1 Δ6nt mouse was generated by CRISPR/ Cas9 injection into the pronucleus of 1-cell zygotes. Guide RNA against exon 3 was used with the following targeting sequence, 5′-gagcaggttacttacgcaa-3′, which was previously used

in Shaw el al. (2017)[8]. Founders were backcrossed with C57BL/6. Genotyping primers are listed in Supplementary Data 1.

**Western blot and antibodies**. Zebrafish embryos at sphere stage were dechorionated using 1 mg/ml pronase and deyolked as previously described[37]. To enrich for nuclear proteins, embryos were lysed in hypotonic lysis buffer (10 mM Tris pH7.5, 10 mM NaCl, 3 mM MgCl2, 0.3% NP40, 10% glycerol, 1x Protease inhibitor) for 10 min on ice and centrifuged 800 g for 8 min. The supernatant containing cytoplasmic fraction was discarded. The pellet containing nuclear proteins was resuspended in MWB (10 mM Tris pH: 7.5, 300 mM NaCl, 4 mM EDTA, 1% NP40, 10% glycerol, TURBO DNase, 1x Protease Inhibitor). Nuclear proteins from 300 embryos were loaded per lane. Human fibroblasts and mouse embryo heads were lysed in RIPA buffer (250 mM Tris, pH: 7.5; 150 mM NaCl; 1% NP-40; 0.5% Na deoxycholate, Halt protease inhibitors [Thermo]). Immunoblotting was performed using the following antibodies: anti-SMCHD1 (Atlas HPA039441, 1:500), anti-Histone H3 (abcam ab1791, 1:1000), anti-GAPDH (Santa Cruz SC47724, 1:1000).

In situ hybridization Probes for in situ hybridisation were cloned from zebrafish embryo cDNA using primers in Supplementary Data 1.

**Zebrafish whole mount immunofluorescence**. 33 hpf embryos were dechorionated and fixed for 2 h at room temperature in 4% paraformaldehyde in PBS. The embryos were stored in methanol at −20 °C and then rehydrated in a gradient of PBS/methanol. The embryos were blocked in PBDT (PBS, 1% BSA, 1% DMSO and 1% Triton X-100) with 5% goat serum at room temperature for 1 h. Embryos were incubated with anti-GnRH antibodies (Sigma G8294, 1:500) overnight at 4 °C and washed in PBDT. Secondary antibody used was Alexa fluorophore-labelled anti-rabbit (1:500, Molecular Probes). Embryos were mounted with 2% low melting agarose on slides and imaged with Olympus FV1000 confocal microscope. Terminal nerve length was measured using ImageJ.

**qPCR**. RNA from 30-50 zebrafish embryos at the designated stages or human fibroblast was extracted using RNeasy Mini Kit (Qiagen) with DNase digestion on column. cDNA was synthesized from 1 μg of total RNA using the iScript cDNA Synthesis Kit (BioRad). qRT-PCR was performed using Power SYBR Green PCR Master Mix (Thermo Fisher) on QuantStudio 7 (Thermo Fisher). Gene expression was normalized to *actin* for zebrafish samples and *HPRT* for human samples. Data represent mean ± SD of biological replicates. A list of primer sequences can be found in Supplementary Data 1.

**miRNA expression analysis**. RNA from 30-50 zebrafish embryos at the designated stages was extracted using Trizol (Thermo Fisher). cDNA was synthesized using Mir-X miRNA First Strand Synthesis kit (Takara Bio) and qPCR performed using TB Green Advantage qPCR Premix (Takara Bio). Gene expression was normalized to *U6* using primers provided with the kit. Data represent mean ± SD of biological replicates. Primer sequences can be found in Supplementary Data 1.

**Skeletal analysis**. Zebrafish of 2 months or older were euthanized in an ice bath. The skin, eyes and internal organs of the fish are removed before fixing in 4% PFA at RT for 2 days. Fish are washed with water and moved into a saturated sodium borate solution for 0.5 days. Fish are then bleached in 0.45% H2O2 in 1% KOH for 1 h under light, then kept in 1% KOH overnight. Bones are stained in 0.1% alizarin red in 1% KOH in the dark overnight at RT. Skeletons are cleared in 0.05% trypsin in 35% saturated sodium borate overnight followed by 1% KOH overnight. Clearing is repeated until bones are clearly seen. Skeletons are passed through a glycerol-1% KOH series and stored in 80% glycerol in 1% KOH.

For craniofacial cartilage staining of larval zebrafish, 5 dpf zebrafish were fixed overnight in 4% PFA then dehydrated through a methanol series. Larval fish are stained overnight in Alcian Blue Stain (0.1% Alcian Blue, 70% Ethanol, 30% Glacial Acetic Acid) followed by 3 washes in 100% Ethanol for 5 min each. Larval fish are then rehydrated through an ethanol series into PBST and digested in 0.05% trypsin for 2 h at room temperature. Tissue is cleared in 1% KOH for 10 min. The skeletons are passed through a glycerol-1% KOH series and stored in 80% glycerol in 1% KOH.

P0 mouse pups were deskinned, eviscerated and fixed in 95% ethanol overnight. They were then transferred to acetone overnight to remove fats. Pups were stained with alcian blue solution (0.05% Alcian Blue, 80% Ethanol, 20% Glacial Acetic Acid) for 24 hours. They were washed in 70% Ethanol over 6-8 h. They were cleared in 1% KOH until the tissues are visibly cleared, then stained in Alizarin red solution (50 ug/ml Alizarin red in 1% KOH) in the dark for 6 h. The skeletons were de-stained in 1% KOH for 1 day. They are then passed through a glycerol-1% KOH series and stored in 80% glycerol in 1% KOH.

**RNA-seq analysis**. 3 biological samples each of *wt* and *MZ lof1* embryos at 4-8-cell and sphere stages were used in the RNA-seq. The quantity and integrity of RNA were measured using the Bioanalyzer 2100 (Agilent Technologies). Samples were poly-A enriched and sequenced (paired-end) on HiSeq4000 (Illumina). Reads were mapped onto the danRer10 zebrafish reference genome and the genes were annotated using the *Danio rerio* annotation from Ensembl release 89. Genes with a normalized mean expression < = 10 and ribosomal RNA genes were removed from further analysis.

DESeq2 was used to perform differential gene expression analysis. Genes with a FDR value below 0.05, mean expression > 10 and fold change > 2 were classified as differentially expressed (DEGs). K-means clustering was performed using silhouette width and within-sum-of-squares metrics. DEGs were enriched against Gene Ontology Biological Processes (GO:BP), Gene Ontology Molecular Function (GO:MF), and Kyoto Encyclopaedia of Genes and Genomes (KEGG) ontologies. Gene Set Enrichment Analysis (GSEA) was performed on all genes expressed. RNA seq methods in primary fibroblast are described in Laberthionniere et al.[38]

**DNA methylation analysis**. Unfertilized oocytes were squeezed out from anesthetized adult female fish. Oocytes were activated in egg water for 10 min. Eggs that failed to activate or had lysed were discarded. Healthy eggs were collected into tubes and lysed in a chilled cell lysis solution (10 mM Tris pH8, 10 mM NaCl, 0.5% NP40) by passing them through a 20 G needle. Nuclei were collected by centrifugation at 3500 g for 5 min at 4 °C. The supernatant was removed and the nuclei pellet lysed in nuclear lysis buffer (50 mM Tris pH8, 10 mM EDTA, 1% SDS, 10 µl Proteinase K, 2 µl RNase A/ml) at 55 °C for 2 h. DNA was extracted by phenol chloroform extraction. DNA was digested again with RNase A at 37 °C for 3 h, followed by phenol chloroform extraction. DNA from 500 eggs was subjected to bisulfite conversion using MethylEdge Bisulfite Conversion System (Promega). PCR was performed with HotStarTaq (Qiagen) and PCR product TA-cloned into pGEM-T Easy (Promega). 16 clones were picked per genotype and sequenced with T7 primer. Efficiency of bisulfite conversion was over 99% as observed by the conversion of non-CpG cytosines.

**Zebrafish odour tests**. In total, 2–4 weeks old larval fish were gently introduced into the observation tank (75 mm × 30 mm x 30 mm (L x W x H) with 40 ml of water. Video recordings started after 3–5 min of acclimation. After 5 min of video recording, 0.5 ml of 100 µM glycochenodeoxycholic acid (GCDA) at one end of the tank and a similar volume of tank water from the other end were delivered. Recording continued for the next 5 minutes. Fish were tracked online and tracked data was analyzed as described previously[39]. In total, 20% area of the tank from the delivery port was defined as the stimulus zone.

*Schreckstoff* preparation and behavioural data analysis was performed as described previously[40] with minor modifications. Briefly, *Schreckstoff* was prepared from euthanized zebrafish by introducing 7 to 10 shallow lesions with a Sharpoint knife (22.5° stab). Fish were then immersed into 2 ml of fish water for 1 to 2 min and rocked on a rocker. The 2 ml crude extract was then centrifuged at 20,000 × *g*, filtered, and heated overnight at 95 °C.

Adult subject fish were transferred from the fish facility and gently introduced into the observation tank. Video recordings started after 5–7 min of acclimation. 250 µl of *Schreckstoff* was introduced via an automated delivery valve after 120 s of video recording (Pre) at 1 ml/minute. Recording continued for the next 120 s including the delivery period (Post). For analyzing the position and speed of the fish swimming, videos were re-digitized at 15 fps and fish position was tracked automatically using the "track objects" algorithm in MetaMorph. 6.3. Automated algorithms were used to compute all parameters described from the XY position data of the subject.

**5-Aza-dC treatment**. 8-cell stage embryos were treated with various concentrations of DNA methylation inhibitor 5-Aza-2′-deoxycytidine (5-Aza-dC) for 48 h then washed to remove the inhibitor. Embryos were observed for survival and developmental defects. Dead embryos were removed daily.

**Cell culture**. Written informed consent of individuals were received prior to collection of samples and subsequent analysis for research purposes according to the ethical approvals of the local Institutional Review Boards in Singapore (A*STAR IRB 2019-087, NUS IRB N-20-054E) and France (La Timone Children's Hospital). All skin biopsies except for one control (foreskin) were taken from the forearms of individuals. Primary fibroblasts were cultured in DMEM (Life Technologies) supplemented with 10% foetal bovine serum (Hyclone). gRNAs targeting SMCHD1 were cloned into LentiCRISPRv2 puro (a gift from Brett Stringer, Addgene plasmid # 98290). Two guide RNAs targeting 5′- CTTACCTCACTATGACACAC- 3′ (KO1) and 5′- TTGTTATTACAACAACAAGT- 3′ (KO2) were used. Lentiviruses were packaged with third generation packaging plasmids. Following viral transduction, cells were selected in 1 µg/ml puromycin.

**Statistics**. All graphs represent means ± standard deviation. *P* values were calculated by the Kruskal-Wallis test followed by Dunn's Multiple Comparison Test, unless otherwise stated. *$p < 0.05$, **$p < 0.01$, ***$p < 0.001$.

**Reporting summary**. Further information on research design is available in the Nature Research Reporting Summary linked to this article.

## Data availability
The RNA-seq data produced in this study were submitted to GEO under the following accession number: GSE173462, GSE174604. Source data are provided as a Source Data file. Source data are provided with this paper.

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

## Acknowledgements

We are grateful to all members of the Reversade and Xue lab for support. We thank Keerthika, Lydia Ng, Glenn Koh for technical assistance, IMCB-MMD (formerly IMB-KOre unit) for generating the knockout mice and P. Ingham for feedback. B.R. is indebted to Davor Solter, for his mentorship, acumen and inspiration. This work was supported by NMRC/OFYIRG/0062/2017 and NUS PYP startup grant to S.X. Work in A.S.M's lab was supported by the Ministry of Education, Singapore (MOE) and Yale-NUS College through IG19-BG106, MOE-T2EP30220-0020, and an SUG. Work in F.M.'s lab was funded by "Association Française contre les Myopathies" (AFM; TRIM-RD, MoThARD) and Fondation Maladies Rares. B.R. is an investigator of the National Research Foundation (NRF, Singapore), Branco Weiss Foundation (Switzerland) and an EMBO Young Investigator, and is supported by an inaugural Use-Inspired Basic Research (UIBR) central fund from the Agency for Science & Technology and Research (A*STAR) in Singapore.

## Author contributions

S.X. and B.R. designed the study and obtained funding. S.X., T.T.N.L., J.C. and J.N. performed the experiments. A.S.M. performed and analyzed olfactory experiments. A.S.M and R.S.V analyzed RNA-seq data. F.M. provided patient material and data. S.X. and B.R. wrote the manuscript with input from all authors.

## Competing interests

The authors declare no competing interests.
