## [Peer Review File · Nature Communications]

HOX epimutations driven by maternal SMCHD1/LRIF1 haploinsufficiency trigger homeotic transformations in genetically wildtype offspringREVIEWER COMMENTS

Reviewer #1 (Remarks to the Author):

In this manuscript Xue et al. produce zebrafish CRISPR/Cas9 mutants of SMCHD1 and LRIF1, two genes involved in imprinting in mammals. In contrast to mice, the homozygous zebrafish mutants are viable but show defects in their axial skeleton, most notably a reduction in vertebral number. Genetic analysis indicates that this phenotype depends on the genotype of the maternal germline and is not rescued by zygotic complementation with the wildtype SMCHD1 gene. RNAseq analysis of 2-4 cell and cell embryos indicates upregulation of several groups of hox genes, of which *hoxb2a* is validated using in situ hybridization and shows a slight posterior expansion in the hindbrain. Additional knockout of the SMCHD1 interacting protein LRIF1 induces a similar axial phenotype. Importantly double SMCHD1 and LRIF1 LOF does not induce more drastic alterations, strongly indicating the genes are active in the same pathway as expected. Additional methylation analysis in zebrafish is performed for *hoxc10*. Furthermore, patient fibroblast samples haploinsufficient for SMCHD1 are investigated which indicate hox misregulation as a result of misregulation during germline formation, since SMCHD1 somatic knockout in does not affect hox expression in wildtype fibroblasts.

I think this is an interesting manuscript and might be considered as a contribution for Nature Communications. However, there are a number of aspects that should be addressed. My detailed comments are listed below but most importantly I think the authors should provide a more detailed analysis of differentially expressed genes in the RNAseq, provide a further analysis and quantification of the hox overexpression phenotype, look further into the genetic basis of the phenotype (and decide if these are truly homeotic transformations) and make absolutely sure that the patient fibroblast data presented cannot be an artefact that results from their sampling location instead of from their mutant genotype.

- Concerning the RNAseq analysis. I think it is important not just to focus on the hox genes but to obtain an unbiased view of which genes are up/downregulated. The authors should perform a GO term analysis to see whether specific gene classes can be identified to be affected. Are these for instance all developmental genes?
- Then concerning the upregulation of the hox genes, I think it is important to know how the observed expression values compare (e.g. FPKM or QPCR values compensated for a housekeeping gene) to the normal expression levels as are observed during the early activation of the hox genes at gastrulation/neurulation. Currently only a comparison is provided for the early zygotic stages when the normal hox program has not yet been activated and therefore it is difficult to compare the levels of precocious activation in the mutants with normal physiologically relevant hox expression levels. It occurs to me that the "leakage" as occurs from misregulation in the germline might be very low, but is this ten fold lower, a hundred fold? I think some number should be put on this.
- A similar concern for the *hoxb2a* expression data. The difference in expression appears very minor and in addition not just the more posterior rhombomere has overexpression but this seems present throughout the embryo, also more anteriorly. In fact, the authors claim that it is the germline expression rather than the zygotic expression that is affected. Therefore, this could be exactly what is expected. I suppose the authors also investigated earlier stages using in situ. How do these look like? Or if not, I think a more broader analysis of earlier stages should be presented.
- Concerning the "homeotic transformations" the authors identify I am not quite convinced that these can be attributed to the function of the hox genes. Homeotic transformations (i.e. the change of segmental identity) normally manifest themselves as a change in vertebral identity rather than a simple reduction in vertebral numbers. Examples in mouse for instance are expansion or reduction of the rib-bearing vertebrae by *hox6* (Carapuço et al. Genes Dev 2005) or *hox10* (Wellik & Capecchi Science 2003) genes or induction of ectopic sacral identity by *hox11* genes (Woltering & Duboule Mech Dev 2015). Also, the hox gene group that is most highly upregulated in the SMCHD1, *hox2*, might be

expected to induce transformations in the pharyngeal skeleton (Pasqualetti et al. Development 2000), which according to supp. Fig. 1a looks normal. Also, effects in the patterning of hindbrain rhombomeres could be expected, did the authors look at the hindbrain neuronal pattern? This can be done quite simple using retrograde labelling, or *islet1/tag1* in situ (see for instance McClintock et al. Development 2002, Woltering & Durston PlosOne 2008).

One *hox* related pathway that springs to mind and that can induce vertebral reductions when overexpressed in the zebrafish egg through micro-injection is the microRNA *mir-196* family (He et al. Dev Biol 2011), which is coexpressed with the associated *hox10* genes. Given that the *hox10* group genes are amongst the most highly upregulated this may provide the biological mechanism for axial reduction. I would strongly encourage the authors to investigate this latter pathway as it could provide the most plausible explanation for their phenotype.

- It is extremely important to know from which region of the body the used fibroblast cell lines were sampled. *Hox* codes have been shown to be maintained in adult fibroblasts according to their anterior posterior position (e.g. Rin et al. Plos Genetics 2006). Therefore, different *hox* codes in different fibroblast cell lines could also be the result of different sampling areas. Can the authors rule out an explanation that different source locations are responsible for the different *hox* codes detected? This should also be documented and justified in the manuscript.

Minor:

- The phrases "Zebrafish has undergone a partial genome duplication and carries seven *hox* gene clusters, instead of four in mammals and one in *Drosophila*1. With so much built-in redundancy, mutations in a single *hox* gene rarely cause overt phenotypes. Surprisingly adult MZ *smchd1lof1/lof1* fish showed patent homeotic transformations (Fig. 2e, f) consisting of a reduction in the number of ribs, caudal vertebrae and occasionally of supraneural vertebrae (Supp. Fig. 3)." is slightly misleading. Indeed, there is a large degree of redundancy amongst the duplicated *hox* genes, however this typically affects *hox*-LOF conditions but is not expected to affect gain of function conditions as reported here. The best would be to rephrase this in context.

- Please provide details concerning the genotyping of the zebrafish lines in the material and method section

Reviewer #2 (Remarks to the Author):

This manuscript explores the role of the evolutionary-conserved chromatin-binding protein SMCHD1 in regulating developmental genes in Zebrafish. SMCHD1 has been shown to have an involvement in X-chromosome inactivation in mice, and more recently in regulated imprinted gene expression, where it has both zygotic and maternal-effect contributions. In human, haploinsufficiency or loss-of-function mutations in SMCHD1 cause facioscapulohumeral dystrophy type 2 (FSHD2) or craniofacial disorders. In this study, the authors derive knock-outs of *smchd1* in Zebrafish. Briefly, they find that:

- zygotic mutants have few gross phenotypic effects;

- maternal-zygotic mutants have subtle homeotic transformations, notably a reduction in vertebra number;

- associated with this phenotype a subset of *hox* genes is de-repressed in unfertilised oocytes and early embryos;

- at one *hox* gene, they correlate de-repression in oocytes with reduction in DNA methylation in oocytes;

- they find that mutations in *Lr1f1*, a known partner of *Smchd1*, phenocopy those of *smchd1* mutations, with no additive effects in double mutants. This is of interest, because *Lr1f1* is apparently only expressed at early stages of oogenesis, indicating a legacy in offspring from early events in the germline.

Critically, they show that *smchd1* is haploinsufficient in zebrafish eggs, such that genetically wild-type offspring from heterozygous females also exhibit homeotic transformations. Finally, they also explore

the effect of SMCHD1 mutation in fibroblasts from FSHD2 patients, finding de-regulation in the pattern of HOX gene expression. Notably, this molecular phenotype depends upon prior (germline?) loss of SMCHD1, because the same effect (de-repression of a subset of HOX genes) cannot be induced in control fibroblasts by genetic inactivation of SMCHD1.

In all, this is a really study, showing *Smchd1* to be a maternal-effect gene that is dosage sensitive in the maternal germline with an intergenerational effect also upon genetically wild-type progeny. The results are presented very concisely and I suggest that the authors need to expand the manuscript in some areas to more fully cover the results or more clearly demonstrate the reproducibility of some of the findings. The major comments I have are as follows.

1. The analysis of SMCHD1 mutant fibroblasts is over-interpreted. The authors state (lines 176-178) "Our results indicate that HOX misexpression caused by SMCHD1 haploinsufficiency is set up in the germline, can be stably inherited through somatic cell divisions and cannot be recapitulated by deleting SMCHD1 post-fertilization." The experiment performed cannot distinguish between an effect of SMCHD1 haploinsufficiency in the female germline or at early embryonic stages when cell-lineage specific epigenomes are being established. This needs to be acknowledged. If there were an effect from the germline, this may also impact genetically normal sibs of FSHD2 cases from female carriers – similar to the observations the authors have made in the zebrafish mutants. Presumably this has been examined in SMCHD1/FSHD2 families.

Similarly, the title of Figure 5 needs to be revised to avoid this over-interpretation, as well as the statement (lines 197-199) "This also invites the question of whether pervasive HOX epi-mutations, driven by insufficient SMCHD1/LRIF1 activity during gametogenesis, may contribute to the pathogenesis of FSHD2 which exhibits anterior-biased muscle degeneration."

2. The DNA methylation analysis (Fig. 3f) of the *hoxc10a* locus is described minimally and needs to be improved to be a convincing result. The authors perform conventional bisulphite sequencing and cloning for methylation analysis of part of the *hoxc10a* gene from oocytes of *smchd1-lof* and control females, stating (lines 378-379) that "16 clones were picked per genotype". How many replicates were done? The number of eggs used (500) is towards the low end for this method, and PCR amplification of bisulphite-converted DNA very prone to clonality issues (as might be suggested from the similarity of many of the methylation profiles shown), so knowing how the authors removed possible clonal products (e.g., from non-conversion events at non-CpG cytosines), and knowing whether similar patterns were obtained in multiple replicate samples is essential. It would also be useful to have a scheme to show where the analysed region is in relation to the gene and promoter.

3. The RNA-seq analysis is also very minimally described. We are not told how many samples per genotype were tested. Although it may be legitimate to focus the narrative on effects on a subset of Hox genes, it is not clear that these genes are among those with the greatest effect. DESeq analysis identifies >2000 differentially expressed genes at the 4-8 cell stage and >4800 at the sphere stage (Suppl. Table 2). I suggest that the authors discuss the RNA-seq results more fully before going on to focus on the Hox genes of interest.

Reviewer #1 (Remarks to the Author):

In this manuscript Xue et al. produce zebrafish CRISPR/Cas9 mutants of SMCHD1 and LRIF1, two genes involved in imprinting in mammals. In contrast to mice, the homozygous zebrafish mutants are viable but show defects in their axial skeleton, most notably a reduction in vertebral number. Genetic analysis indicates that this phenotype depends on the genotype of the maternal germline and is not rescued by zygotic complementation with the wildtype SMCHD1 gene. RNAseq analysis of 2-4 cell and cell embryos indicates upregulation of several groups of hox genes, of which *hoxb2a* is validated using in situ hybridization and shows a slight posterior expansion in the hindbrain. Additional knockout of the SMCHD1 interacting protein LRIF1 induces a similar axial phenotype. Importantly double SMCHD1 and LRIF1 LOF does not induce more drastic alterations, strongly indicating the genes are active in the same pathway as expected. Additional methylation analysis in zebrafish is performed for *hoxc10*.

Furthermore, patient fibroblast samples haploinsufficient for SMCHD1 are investigated which indicate hox misregulation as a result of misregulation during germline formation, since SMCHD1 somatic knockout in does not affect hox expression in wildtype fibroblasts.

I think this is an interesting manuscript and might be considered as a contribution for Nature Communications. However, there are a number of aspects that should be addressed. My detailed comments are listed below but most importantly I think the authors should provide a more detailed analysis of differentially expressed genes in the RNAseq, provide a further analysis and quantification of the hox overexpression phenotype, look further into the genetic basis of the phenotype (and decide if these are truly homeotic transformations) and make absolutely sure that the patient fibroblast data presented cannot be an artefact that results from their sampling location instead of from their mutant genotype.

- Concerning the RNAseq analysis. I think it is important not just to focus on the hox genes but to obtain an unbiased view of which genes are up/downregulated. The authors should perform a GOterm analysis to see whether specific gene classes can be identified to be affected. Are these for instance all developmental genes?

We have performed a GO analysis on the RNAseq data and added the data to the manuscript. At the 4- to 8-cell stage, GO analysis revealed skeletal system development as one of the top categories (**Supp. Fig. 2a-d**). Further clustering of the differentially expressed genes at the 4- to 8-cell stage disclosed 2 clusters of genes which were upregulated in the mutant embryos (**Fig. 2a**). Among these, cluster 5 shows a significant enrichment of genes involved in skeletal system development (**Fig. 2b**). GO analysis at sphere stage did not reveal any significant categories (**Supp. Fig. 2a-d**).

- Then concerning the upregulation of the hox genes, I think it is important to know how the observed expression values compare (e.g. FPKM or QPCR values compensated for a housekeeping gene) to the normal expression levels as are observed during the early activation of the hox genes at gastrulation/neurulation. Currently only a comparison is provided for the early zygotic stages when the normal hox program has not yet been activated and therefore it is difficult to compare the levels of precocious activation in the mutants with normal physiologically relevant hox expression levels. It occurs to me that the “leakage” as occurs from misregulation in the germline might be very low, but is this ten fold lower, a hundred fold? I think some number should be put on this.

We have included qPCR of some *hox* genes over a developmental time course (**Supp Fig 3a**). The reviewer is right that early activation levels are much lower than the normal expression levels at somitogenesis. However, for most *hox* genes, we observe persistent upregulation of the *hox* expression in *smchd1* null mutants throughout embryogenesis, even after somitogenesis, supporting our theory that early loss of *Smchd1* can cause a long-lasting change in *hox* expression. We also note from this data that there isn't a change in the timing of zygotic *hox* expression as the peak of *hox* expression remains at somitogenesis.

- A similar concern for the *hoxb2a* expression data. The difference in expression appears very minor and in addition not just the more posterior rhombomere has overexpression but this seems present throughout the embryo, also more anteriorly. In fact, the authors claim that it is the germline expression rather than the zygotic expression that is affected. Therefore, this could be exactly what is expected. I suppose the authors also investigated earlier stages using in situ. How do these look like? Or if not, I think a more broader analysis of earlier stages should be presented.

Unfortunately, while we could document an increase in *hoxb2a* expression by RNA-seq and qPCR, the sensitivity of in situ hybridisation was not high enough to convincingly detect a signal of *hoxb2a* at the 8-cell stage in either the wildtype or the mutant (**Supp Fig 3b**).

To show that the expansion of expression boundary is not restricted to one *hox* gene, we performed in situ hybridisation of another *hox* gene, *hoxc10a*, at 17-somites. We show that it too has a pronounced expansion of the expression boundary (**Fig 2f**).

- Concerning the “homeotic transformations” the authors identify I am not quite convinced that these can be attributed to the function of the *hox* genes. Homeotic transformations (i.e. the change of segmental identity) normally manifest themselves as a change in vertebral identity rather than a simple reduction in vertebral numbers. Examples in mouse for instance are expansion or reduction of the rib-bearing vertebrae by *hox6* (Carapuço et al. *Genes Dev* 2005) or *hox10* (Wellik & Capecchi *Science* 2003) genes or induction of ectopic sacral identity by *hox11* genes (Woltering & Duboule *Mech Dev* 2015). Also, the *hox* gene group that is most highly upregulated in the SMCHD1, *hox2*, might be expected to induce transformations in the pharyngeal skeleton (Pasqualetti et al. *Development* 2000), which according to supp. Fig. 1a looks normal. Also, effects in the patterning of hindbrain rhombomeres could be expected, did the authors look at the hindbrain neuronal pattern? This can be done

quite simple using retrograde labelling, or *islet1/tag1* in situ (see for instance McClintock et al. *Development* 2002, Woltering & Durston *PlosOne* 2008).

Following your suggestions, we performed *islet1* in situ in our embryos to look at motor neuron patterning in the hindbrain. While the intensity of the staining is slightly different between the *wt* and *MZlof* embryos, we did not observe a major patterning change (**Supp Fig 3f**). We understand the concern with calling the vertebral reductions “homeotic transformations”, and have changed the wording to call them vertebral patterning defects.

In addition, we have new mouse data that supports the regulation of *Hox* by SMCHD1. We generated a new *Smchd1* knockout allele in C57BL/6 mice by CRISPR/Cas9. Similar to previously generated *Smchd1* knockout mice (*Smchd1^{MommeD1/MommeD1}*)¹, homozygous females were embryonic lethal while there was also a sublethal phenotype in males (**Fig. 4c**). Previous studies have shown that SMCHD1 regulates long range chromatin interactions at

Hox clusters². Axial skeletal anomalies were seen in our newly created heterozygous *Smchd1*^{Δ^{mt}} mice whereas they have only been reported in homozygous knockout males previously².

The observed anomalies were classical homeotic transformations and were consistent with those previously described including an ectopic rib at C7 (C7->T1) and absence of ribs at T13 (T13->L1). Additionally, consistent transformation of L6 to S1 was observed (**Fig. 4d, e**). We were unable to ascertain whether these observed axial transformations were due to a maternal or zygotic loss of gene activity as all *Smchd1* knockout females were embryonic lethal. However, this conundrum has been recently addressed by Benetti and colleagues³(bioRxiv, also in review at Nature Communications) who have shown, using an oocyte-specific Cre-mediated deletion of maternal of *Smchd1*, that the maternal pool of *Smchd1* has an important role in repressing Hox expression and that it also leads to vertebral patterning defects in pups born to mothers only lacking *Smchd1* during oogenesis. These results are consistent with those made in zebrafish and confirm that *Smchd1* is essential in the maternal germline, before fertilization, to control the expression of Hox genes in future embryos.

We have added this replicative but mammalian-centric data as a new **Figure 4**.

One hox related pathway that springs to mind and that can induce vertebral reductions when overexpressed in the zebrafish egg through micro-injection is the microRNA mir-196 family (He et al. Dev Biol 2011), which is coexpressed with the associated hox10 genes. Given that the hox10 group genes are amongst the most highly upregulated this may provide the biological mechanism for axial reduction. I would strongly encourage the authors to investigate this latter pathway as it could provide the most plausible explanation for their phenotype.

We analysed the abundance of the mature mir-196 family members by qPCR. Expression of mature mir196 paralogs was not significantly changed in *smchd1*^{lofl} oocytes but was increased approximately 1.5 fold at 11-somites, suggesting that it may also participate in the general dysregulation of the *hox* transcripts (**Supp. Fig. 3d, e**). Other mir196-related phenotypes such as the loss of pectoral fins were not observed. As mir196-induced vertebral reductions were previously achieved with an estimated 10-20 fold mir196 over-expression, we surmise that the modest mir196 upregulation may not be a main driver of the phenotype in our *smchd1* knockout fish.

Most *hox* studies have been done with knockouts. Studies have shown that overexpression and knockouts of *hox* genes do not always give opposite phenotypes⁴. We suggest that the phenotype we observe may be the result of upregulation of multiple *hox* genes and cannot be attributed to the roles of individual *hox*. In accordance with this, other models that change total vertebral numbers include Mir-196 + Gdf11 which together regulate the expression of multiple Hox genes at the same time^{5,6}, and overexpression of a whole cluster of *hox*⁷.

- It is extremely important to know from which region of the body the used fibroblast cell lines were sampled. Hox codes have been shown to be maintained in adult fibroblasts according to their anterior posterior position (e.g. Rin et al. Plos Genetics 2006). Therefore, different hox codes in different fibroblast cell lines could also be the result of different sampling areas. Can the authors rule out an explanation that different source locations are responsible for the different hox codes detected? This should also be documented and justified in the manuscript.

All skin biopsies except for one control (foreskin) were taken from the forearms of individuals. This has been added to the methods. There isn't much difference in *HOX* gene expression between the two controls that we used (**Fig 6a**), therefore it is unlikely that the differences in *HOX* expression between control and FSHD2 cells are due to the origin of fibroblasts.

Minor:

- The phrases "Zebrafish has undergone a partial genome duplication and carries seven hox gene clusters, instead of four in mammals and one in *Drosophila*1. With so much built-in redundancy, mutations in a single hox gene rarely cause overt phenotypes. Surprisingly adult MZ *smchd1lof1/lof1* fish showed patent homeotic transformations (Fig. 2e, f) consisting of a reduction in the number of ribs, caudal vertebrae and occasionally of supraneural vertebrae (Supp. Fig. 3). " is slightly misleading. Indeed, there is a large degree of redundancy amongst the duplicated hox genes, however this typically affects hox-LOF conditions but is not expected to affect gain of function conditions as reported here. The best would be to rephrase this in context.

We have removed the point about redundancy.

- Please provide details concerning the genotyping of the zebrafish lines in the material and method section

Sorry for the oversight, we have added this to the methods.

Reviewer #2 (Remarks to the Author):

This manuscript explores the role of the evolutionary-conserved chromatin-binding protein SMCHD1 in regulating developmental genes in Zebrafish. SMCHD1 has been shown to have an involvement in X-chromosome inactivation in mice, and more recently in regulated imprinted gene expression, where it has both zygotic and maternal-effect contributions. In human, haploinsufficiency or loss-of-function mutations in SMCHD1 cause facioscapulohumeral dystrophy type 2 (FSHD2) or craniofacial disorders. In this study, the authors derive knock-outs of *smchd1* in Zebrafish. Briefly, they find that:

- zygotic mutants have few gross phenotypic effects;
- maternal-zygotic mutants have subtle homeotic transformations, notably a reduction in vertebra number;
- associated with this phenotype a subset of hox genes is de-repressed in unfertilised oocytes and early embryos;
- at one hox gene, they correlate de-repression in oocytes with reduction in DNA methylation in oocytes;
- they find that mutations in *Lrif1*, a known partner of *Smchd1*, phenocopy those of *smchd1* mutations, with no additive effects in double mutants. This is of interest, because *Lrif1* is apparently only expressed at early stages of oogenesis, indicating a legacy in offspring from early events in the germline.

Critically, they show that *smchd1* is haploinsufficient in zebrafish eggs, such that genetically wild-type offspring from heterozygous females also exhibit homeotic transformations. Finally, they also explore the effect of SMCHD1 mutation in fibroblasts from FSHD2 patients, finding de-regulation in the pattern of HOX gene expression. Notably, this molecular

phenotype depends upon prior (germline?) loss of SMCHD1, because the same effect (depression of a subset of HOX genes) cannot be induced in control fibroblasts by genetic inactivation of SMCHD1.

In all, this is an really study, showing *Smchd1* to be a maternal-effect gene that is dosage sensitive in the maternal germline with an intergenerational effect also upon genetically wild-type progeny. The results are presented very concisely and I suggest that the authors need to expand the manuscript in some areas to more fully cover the results or more clearly demonstrate the reproducibility of some of the findings. The major comments I have are as follows.

1. The analysis of SMCHD1 mutant fibroblasts is over-interpreted. The authors state (lines 176-178) “Our results indicate that HOX misexpression caused by SMCHD1 haploinsufficiency is set up in the germline, can be stably inherited through somatic cell divisions and cannot be recapitulated by deleting SMCHD1 post-fertilization.” The experiment performed cannot distinguish between an effect of SMCHD1 haploinsufficiency in the female germline or at early embryonic stages when cell-lineage specific epigenomes are being established. This needs to be acknowledged. If there were an effect from the germline, this may also impact genetically normal sibs of FSHD2 cases from female carriers – similar to the observations the authors have made in the zebrafish mutants. Presumably this has been examined in SMCHD1/FSHD2 families.

Similarly, the title of Figure 5 needs to be revised to avoid this over-interpretation, as well as the statement (lines 197-199) “This also invites the question of whether pervasive HOX epimutations, driven by insufficient SMCHD1/LRIF1 activity during gametogenesis, may contribute to the pathogenesis of FSHD2 which exhibits anterior-biased muscle degeneration.”

Thank you for pointing this out. While we do not have evidence on whether genetically normal siblings of FSHD2 cases show clinical phenotypes, our data in zebrafish and another group’s independent data in mouse³ (biorXiv, also in review at Nature Communications) show the Hox misregulation is caused by germline *Smchd1* inactivation. We agree with the reviewer that our fibroblast experiment cannot distinguish between SMCHD1 haploinsufficiency in germline or early embryonic stages. Therefore we have updated the title of Figure 6 (previously Figure 5) to “Germline, but not adult, SMCHD1 deficiency leads to HOX dysregulation in human cutaneous fibroblasts.” We also included early embryogenesis in the text as a possible explanation for the fibroblast data.

The sentence at lines 197-199 is placed in the discussion where authors are expected to ponder about the meaning and possible implications of their findings. It is intentionally formulated as a hypothesis (and not an interpretation) using carefully chosen words to convey uncertainty: “invites the question” and “may contribute” to discuss the pathogenesis of FSHD2. We have chosen to leave this sentence in.

Notably, hypoplastic or absent ribs have been reported in some BAMS patients carrying *de novo* missense mutations in SMCHD1⁸, drawing a possible link between SMCHD1 and anterior-posterior patterning in humans.

2. The DNA methylation analysis (Fig. 3f) of the *hoxc10a* locus is described minimally and needs to be improved to be a convincing result. The authors perform conventional bisulphite sequencing and cloning for methylation analysis of part of the *hoxc10a* gene from oocytes of

smchd1-lof and control females, stating (lines 378-379) that “16 clones were picked per genotype”. How many replicates were done? The number of eggs used (500) is towards the low end for this method, and PCR amplification of bisulphite-converted DNA very prone to clonality issues (as might be suggested from the similarity of many of the methylation profiles shown), so knowing how the authors removed possible clonal products (e.g., from non-conversion events at non-CpG cytosines), and knowing whether similar patterns were obtained in multiple replicate samples is essential. It would also be useful to have a scheme to show where the analysed region is in relation to the gene and promoter.

We have done 2 biological replicates of *wt* and 3 biological replicates of the *smchd1-lof*. The additional replicates are shown in **Supp Fig 5d**. Efficiency of bisulfite conversion was over 99% as observed by the conversion of the non-CpG cytosines. We have also included bisulphite sequencing of a region adjacent to the *hoxC* locus as a control. DNA methylation of this region does not change during early embryogenesis and we show that it does not change between *wt* and *smchd1-lof* oocytes as well (**Supp Fig 5e**). The methylation profile in this region looks less clonal and serves to show that our methodology is able to pick up differences in methylation. We have added a schematic of the locus with published DNA methylation profiles as a reference (**Supp Fig 5c**).

3. The RNA-seq analysis is also very minimally described. We are not told how many samples per genotype were tested. Although it may be legitimate to focus the narrative on effects on a subset of Hox genes, it is not clear that these genes are among those with the greatest effect. DESeq analysis identifies >2000 differentially expressed genes at the 4-8 cell stage and >4800 at the sphere stage (Suppl. Table 2). I suggest that the authors discuss the RNA-seq results more fully before going on to focus on the Hox genes of interest.

This is a similar point that was made by Reviewer 1. We have now added GO analysis of the RNA-seq which are in **Fig 2** and **Supp Fig 2**. We have also updated the **Supp Table 2** to show only significantly deregulated genes. The RNA seq methods have been updated to include more details, including the number of samples per genotype used (3 per genotype).

References

1. Blewitt, M. E. *et al.* SmcHD1, containing a structural-maintenance-of-chromosomes hinge domain, has a critical role in X inactivation. *Nat. Genet.* **40**, 663–669 (2008).
2. Jansz, N. *et al.* Smchd1 regulates long-range chromatin interactions on the inactive X chromosome and at Hox clusters. *Nat. Struct. Mol. Biol.* **25**, (2018).
3. Benetti, N. *et al.* Maternal SMCHD1 regulates Hox gene expression and patterning in the mouse embryo. *bioRxiv* 2021.09.08.459528 (2021) doi:10.1101/2021.09.08.459528.
4. Ye, Z. & Kimelman, D. Hox13 genes are required for mesoderm formation and axis elongation during early zebrafish development. *Dev.* **147**, (2020).
5. Wong, S. F. L. *et al.* Independent regulation of vertebral number and vertebral identity by microRNA-196 paralogs. *Proc. Natl. Acad. Sci. U. S. A.* **112**, E4884–E4893 (2015).
6. Hauswirth, G. M. *et al.* Breaking constraint of mammalian axial formulae. *Nat. Commun.* **2022** *131* **13**, 1–12 (2022).
7. Woltering, J. M. & Duboule, D. Tetrapod axial evolution and developmental constraints; Empirical underpinning by a mouse model. *Mech. Dev.* **138**, 64–72 (2015).

8. Delaney, A. *et al.* Insight into the ontogeny of GnRH neurons from patients born without a nose. *J. Clin. Endocrinol. Metab.* **105**, 1538–1551 (2020).

REVIEWERS' COMMENTS

Reviewer #1 (Remarks to the Author):

The authors have done an extensive and tremendous job in revising their manuscript and addressed all my concerns. I now recommend the manuscript for publication in Nature Communications.